# A strict experimental test of macroscopic realism in a superconducting flux qubit

George C. Knee[1,*,†], Kosuke Kakuyanagi[1,*], Mao-Chuang Yeh[2,*], Yuichiro Matsuzaki[1], Hiraku Toida[1], Hiroshi Yamaguchi[1], Shiro Saito[1], Anthony J. Leggett[2] & William J. Munro[1]

Macroscopic realism is the name for a class of modifications to quantum theory that allow macroscopic objects to be described in a measurement-independent manner, while largely preserving a fully quantum mechanical description of the microscopic world. Objective collapse theories are examples which aim to solve the quantum measurement problem through modified dynamical laws. Whether such theories describe nature, however, is not known. Here we describe and implement an experimental protocol capable of constraining theories of this class, that is more noise tolerant and conceptually transparent than the original Leggett–Garg test. We implement the protocol in a superconducting flux qubit, and rule out (by $\sim 84$ s.d.) those theories which would deny coherent superpositions of 170 nA currents over a $\sim 10$ ns timescale. Further, we address the 'clumsiness loophole' by determining classical disturbance with control experiments. Our results constitute strong evidence for the superposition of states of nontrivial macroscopic distinctness.

[1] NTT Basic Research Laboratories, NTT Corporation, 3-1 Morinosato-Wakamiya, Atsugi, Kanagawa 243-0198, Japan. [2] Department of Physics, University of Illinois at Urbana-Champaign, Urbana, Illinois 61801, USA. * These authors contributed equally to this work. † Present address: Department of Physics, University of Warwick, Gibbet Hill Road, Coventry CV4 7AL, UK. Correspondence and requests for materials should be addressed to G.C.K. (email: gk@physics.org) or to W.J.M. (email: william.munro@lab.ntt.co.jp).

In their original paper, Leggett and Garg (LG) asked whether the flux trapped in a superconducting ring was really 'there' when nobody looks[1]. The systems LG had in mind are micrometre scale loops of superconducting material interrupted with one or more nonlinear elements known as Josephson junctions. Such circuits define two possible states of magnetic flux threading the loop, and modern variants[2] are among the most macroscopic candidates for a quantum bit (or qubit), the basic constituent of various proffered quantum-enhanced technologies such as the quantum computer. When a measurement is made, the qubit is found in one of the two possible states $|g\rangle$ or $|e\rangle$ with a probability that oscillates in time. Observation of such so-called 'Rabi oscillations' is consistent with a textbook quantum mechanical prediction (which generally ascribes a nonzero complex amplitude to each of the states), but not necessarily inconsistent with a classical 'value-definite' description (which prescribes that the system is in exactly one state at any given moment)[3]. The decay envelope of the Rabi oscillations is given by an empirical parameter $T_2$. Huge efforts have been invested in extending this characteristic 'coherence time' to the current state-of-the-art value of $85\,\mu s$ (ref. 4), with a view to crossing the quantum error-correction thresholds and enabling large-scale quantum computation[5]. The guiding question of LG's approach extends beyond their prototypical system: is there a fundamental mechanism preventing macroscopic superpositions from persisting, or is the problem merely about resources? LG's name for the former position is 'macroscopic realism', or 'macrorealism' for short: objective collapse models such as Ghirardi, Rimini, Weber and Pearle (GRWP) theory[6,7] or Penrose's gravitationally induced collapse theory[8] are specific examples which might make the quantum-classical divide well defined.

Motivated by the need for a strict test which could rule out this worldview, LG considered $Q_1$, $Q_2$, $Q_3$ as the value taken by a macroscopic observable $Q$ measured at three consecutive times $t_1$, $t_2$, $t_3$, respectively. LG made the assumption of 'macrorealism per se' (MRPS): that these variables can each be assigned a value $\pm 1$ at all times. Then the constraint[1]

$$Q_1 Q_2 + Q_1 Q_3 + Q_2 Q_3 \geq -1 \qquad (1)$$

will hold. An elementary consequence is

$$\langle Q_1 Q_2\rangle_{\mathrm{G}} + \langle Q_1 Q_3\rangle_{\mathrm{G}} + \langle Q_2 Q_3\rangle_{\mathrm{G}} \geq -1 \qquad (2)$$

where $\langle\ldots\rangle_{\mathrm{G}}$ denotes the average over a 'grand' ensemble (or experimental arrangement) where all three observables ($Q_1$, $Q_2$, $Q_3$) are measured. LG conjoined a premise they termed 'non-invasive measurability' (NIM) to reach

$$\mathrm{LGI}: \quad \langle Q_1 Q_2\rangle_{\bar{3}} + \langle Q_1 Q_3\rangle_{\bar{2}} + \langle Q_2 Q_3\rangle_{\bar{1}} \geq -1, \qquad (3)$$

the Leggett-Garg inequality (LGI). Here, $\langle\ldots\rangle_{\bar{i}}$ (for $i = 1, 2, 3$) denotes an average over a ensemble identical to the grand ensemble, with the exception that the observable $Q_i$ is not measured. If NIM is taken to mean that a suitably careful measurement has no effect on the statistics of measurement outcomes before or after it, it is effectively the premise $\langle\ldots\rangle_{\bar{3}} = \langle\ldots\rangle_{\bar{2}} = \langle\ldots\rangle_{\bar{1}} = \langle\ldots\rangle_{\mathrm{G}}$: that the three ensembles in which experiments are actually performed (see Fig. 1a) are equivalent to the grand ensemble. Here, we include the impossibility of backwards causation (sometimes called Induction[9]) in NIM. When 'shuffling' operations $S_1$ and $S_2$ (which induce coherent oscillations between the two states of interest) intervene respectively between $t_1$ and $t_2$, and between $t_2$ and $t_3$, LGI can be violated by a quantum mechanical system. If the system is sufficiently large (super-critically macroscopic), on the other hand, macrorealism predicts that no such violation is possible.

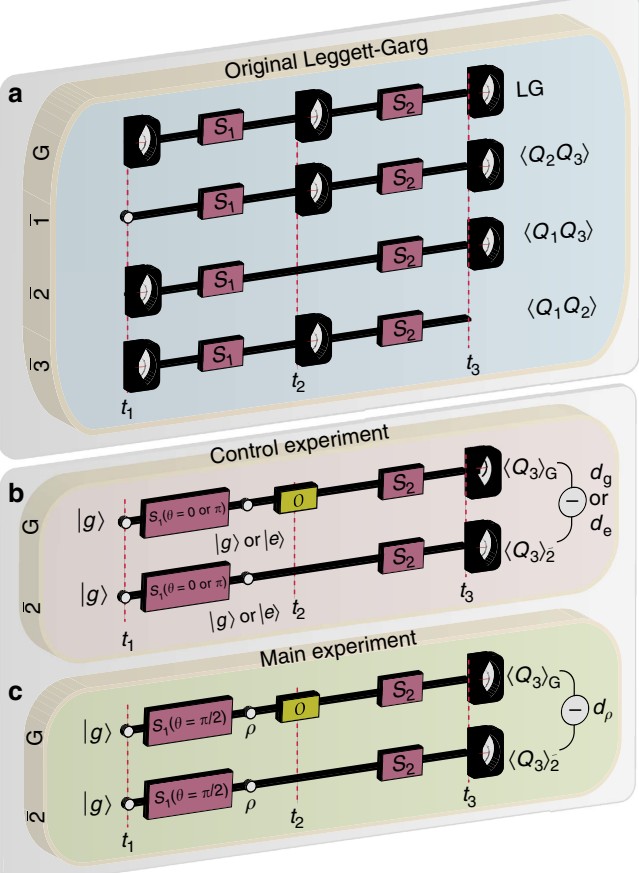

**Figure 1 | A simplified test of macrorealism.** (**a**) Leggett–Garg (LG) derive their inequality-constraint on macrorealism by considering a measurement of a bivalent observable $Q$ at three consecutive times (on an ensemble 'G' of two-level systems). The inequality is tested by gathering two-time correlators from separate experiments, each with a measurement omitted at one of the instants (ensemble '$\bar{1}$' and so on). The lower pane shows our full, simplified experimental protocol. In our experiment the shuffling operations are pulses induced with resonant microwave radiation that cause a pseudo-spin rotation by a variable angle $\theta$, creating coherent superpositions of $|g\rangle$ and $|e\rangle$. The expectation value of a final measurement (at $t_3$) may be influenced by the presence or absence of an earlier operation $O$ (at $t_2$). (**b**) Control experiments determine the worst case disturbance when classical states are prepared: $|e\rangle$ is prepared from the thermal equilibrium state $|g\rangle$ with a $\theta = \pi$ operation. (**c**) The main experiment is identical, only a maximally coherent superposition is prepared with a $\theta = \pi/2$ operation. This gives rise to a measurement disturbance not explainable merely by appealing to the clumsiness revealed in the control experiments.

LGI or variants thereof have been experimentally tested (and violated) in a wide variety of microscopic experimental systems, sometimes with one or more of a variety of additional assumptions. A review of these experiments may be found in ref. 10, but see also more recent experimental tests on a caesium atom[11], delocalized photoexcitations[12] and a two-transmon system[13]. The demanding nature of LG tests has so far influenced the slow progress of experiments toward larger objects, meaning that experiments performed to date at best place only loose bounds on the critical macroscopicity.

Here, we show that LG's approach can be significantly streamlined, resulting in a conceptually cleaner and experimentally simplified protocol. We go on to apply our new protocol to a

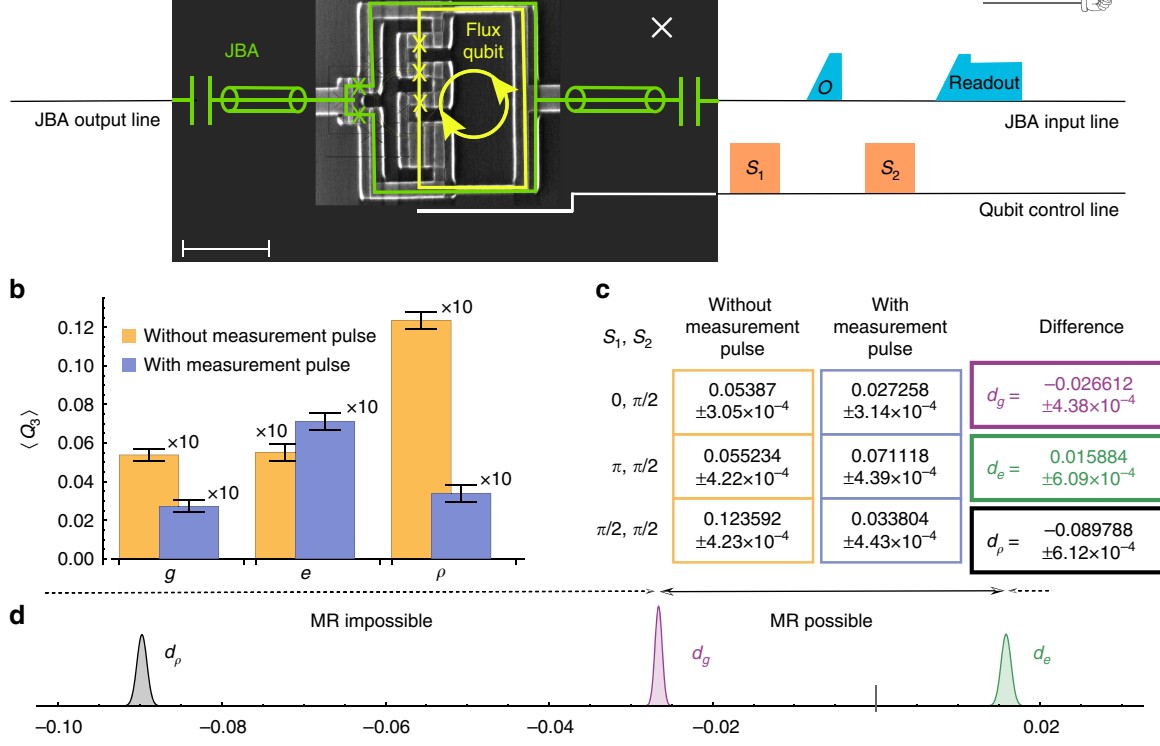

**Figure 2 | Experimental setup and results.** (**a**) Our superconducting flux qubit (yellow) is a micrometre scale device, features three Josephson junctions (crosses), is controlled with microwave pulses (orange) and is readout through magnetic coupling to a Josephson Bifurcation Amplifier (JBA, green) driven by a modulated voltage (blue). Pulses not to scale. (**b**) The expectation value of the final measurement with and without the measurement pulse $O$, for the three preparations $g,e,$ and $\rho$ (represented by a $S_1$ pulse angle of $0,\pi$ and $\pi/2$ respectively, see Fig. 1). The error bars represent 10 s.d. (**c**) The difference between the expectation values is our measure of the disturbance of the operation at $t_2$. The measure is more negative for a quantum preparation ($\rho$) than for either classical preparation ($g$ or $e$). (**d**) $d_\rho$ violates the macrorealist (MR) bound with a high degree of statistical significance.

superconducting flux qubit, thereby pushing the envelope of macroscopicity. The experimental results place constraints on all possible macroscopic-realist theories, and should spur progress towards tests at higher levels of macroscopicity.

## Results

**A simpler test.** MRPS and NIM allow one to reach a simpler constraint

$$\langle Q_3\rangle_G = \langle Q_3\rangle_{\bar{2}}; \quad \text{or} \tag{4}$$

$$\text{NDC}: \quad d := \langle Q_3\rangle_G - \langle Q_3\rangle_{\bar{2}} = 0 \tag{5}$$

with the ensembles as defined previously. We call this equality the 'non-disturbance condition' (NDC). This condition has been derived by others and has been termed a quantum witness[14], 'non disturbing measurement'[15,16] or 'no signalling in time' condition[17]. It follows from the same assumptions as LGI (Supplementary Note 1 and Supplementary Table 1) and demands a zero effect of the choice of measurement at $t_2$ on the statistics of a measurement at the later time $t_3$. Here we suggest, however, that the requirements for the measurement at $t_2$ are very minimal—it will be clear shortly that it need not even be a measurement at all, but some generalized operation $O$ about which we need not assume anything. All pertinent properties of $O$ are to be obtained through experiment.

Inspection of the NDC exposes a number of advantages over LGI. First, that there is no need to measure at all at $t_1$. Second, that only one-point averages, rather than two-point correlations are required. Third, that the condition is an equality rather

than an inequality[18]. The latter two points imply that any non-zero measurement visibility $V := (\max\langle Q\rangle_{\text{obs}} - \min\langle Q\rangle_{\text{obs}})/(\max\langle Q\rangle_{\text{ideal}} - \min\langle Q\rangle_{\text{ideal}})$ (relating observed statistics to ideal ones) is sufficient to find a violation in principle, whereas previously $V > \sqrt{2/3}$ was required.

**Measurement invasiveness.** The ever-present possibility of a clumsy measurement procedure at $t_2$, giving rise to a violation, rather than any inherent quantum effect, is as important a loophole here as ever. The issue has so far only been addressed with *a priori* arguments—those from the use of null result measurements[11,19], weak measurements[20–22] or the use of an additional 'stationarity' assumption[23]. By contrast, Leggett[24] and later Wilde and Mizel[25] have proposed that the problem can be attacked experimentally. This is precisely the approach we adopt here: The classical disturbance of a measurement (which we define shortly) may be determined in a control experiment, rather than assumed zero.

Building on these ideas, here we lay out a precise and operational notion of macrorealism that may be tested in the laboratory. Using conditional probabilities, define the disturbance parameter $d_\sigma := [P(Q_3 = +1|\sigma, O) - P(Q_3 = -1|\sigma, O)] - [P(Q_3 = +1|\sigma) - P(Q_3 = -1|\sigma)]$ as a measure of how much disturbance is introduced to $Q_3$ by applying $O$ at $t_2$ (compared with doing nothing) when the preparation of the system immediately before $t_2$ is described by $\sigma$. In a pair of control experiments, determine $d_g$ and $d_e$, where $g$ and $e$ are the states that the measurement reveals reliably (that is, with 100% chance). These are measures of classical disturbance: see Fig. 1b.

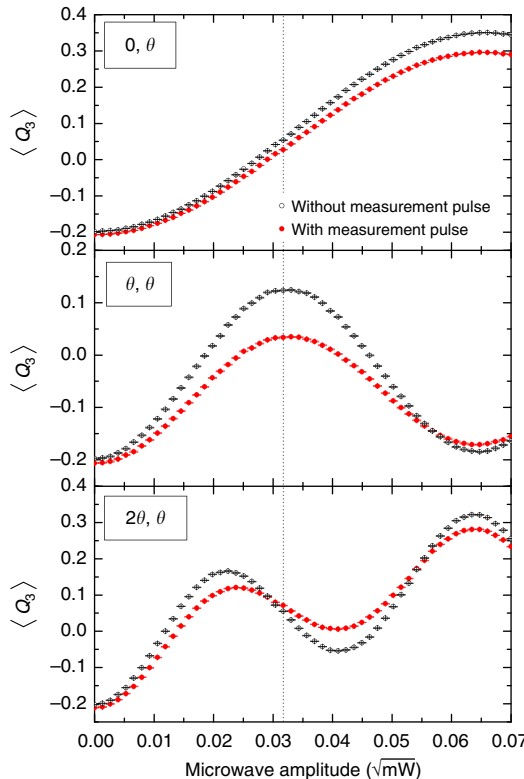

**Figure 3 | Control pulses.** To measure $d_g$, $d_\rho$, $d_e$, the control circuitry of our system was arranged in three separate experiments so that the pseudo-spin rotation angles of $S_1$ and $S_2$ were, respectively $(0, \theta),(\theta, \theta),(2\theta, \theta)$ for a range of angles $\theta$. We then selected the data corresponding to $\theta = \pi/2$, (c.f. Fig. 1b,c) which corresponds to a certain microwave pulse power (shown with a dotted vertical line). Error bars correspond to a 95% confidence interval. The microwave amplitude shown is at the signal generator (it is attenuated before reaching the flux qubit). For quantum mechanical curves, see Supplementary Note 2 and Supplementary Fig. 1.

Once the control experiments are completed, the main experiment may begin to determine $d_\rho$. $\rho$ describes the net preparation when $g$ is followed by a shuffling operation $S_1$ (see Fig. 1c). According to quantum mechanics, the preparation $\rho$ is described by a density operator $\rho = |\alpha|^2 \rho_g + |\beta|^2 \rho_e + \mathcal{C}$. Here, $\rho_{g,e} = |g, e\rangle\langle g, e|$ are the density operators associated with preparations $g$, $e$ respectively, $\alpha$ and $\beta$ are complex numbers satisfying $|\alpha|^2 + |\beta|^2 = 1$, and $\mathcal{C} = (\alpha\beta^\dagger |g\rangle\langle e| + \text{h.c.})$ are off-diagonal 'coherence' terms. In this language the predictions of a macrorealist theory (for super-critically macroscopic systems) are equivalent to those which follow from putting $\mathcal{C} = 0$.

Under the assumption that $S_1$ merely prepares weighted convex combinations (statistical mixtures) of the preparations associated with $g$ and $e$, and does not affect the operation of the measurement at $t_2$, it could be thought natural that the disturbance $d_\rho$ should not be higher than the similarly weighted linear combination of each individual disturbance:

$$d_\rho = |\alpha|^2 d_g + |\beta|^2 d_e, \qquad (6)$$

or even the much weaker

$$\min(d_g, d_e) \leq d_\rho \leq \max(d_g, d_e) \qquad (7)$$

to cover the possibility that the shuffling operation simply deterministically prepares the worst state (that is, the one with the highest susceptibility to disturbance by $O$). The fact that (theoretically at least) $d_g = d_e = 0$ but $d_\rho \neq 0$ could be thought of

as an instance of 'super-activation'[26]. Our definition of macrorealism (7) is a noise-tolerant version of Maroney and Timpson's 'operational eigenstate mixture macrorealism'[16].

If the quantum disturbance $d_\rho$ is significantly greater in magnitude than the greatest classical disturbance, this implies that the shuffling operation prepared something other than a statistical mixture of $g$ and $e$. On the quantum view, this would be a coherent superposition of the preparations. On a hidden-variable view, where preparation of a pure quantum state is actually a stochastic selection from a set of underlying states of reality $\{\lambda_i\}$, $\rho$ might access a new set of $\{\lambda_i\}$ not selectable via either $e$ or $g$; or indeed represent a distribution over the same $\{\lambda_i\}$ that is further from equilibrium with respect to $O$ (ref. 16). It is worth noting that the leading theories of macrorealism do not rely on such hidden states, and so (along with a whole class of future theories subscribing to (7)) can indeed be ruled out by our approach.

**Experimental results.** Now, let us test the protocol experimentally using a superconducting flux qubit, where $O$ is a measurement whose result is not inspected: Schild and Emary[27] call this a 'blind measurement' but here we refer to it as a 'measurement pulse' due to the way it is implemented in our system. We find $d_e > d_g$, and a violation of (7) $d_\rho - d_g = -0.063177$ which is $\sim 84$ s.d. away from zero (shown in Fig. 2). Despite our use of a relatively low fidelity qubit, we are able to reach very low uncertainties by performing $7 \times 10^6$ trials per data point (see Fig. 3). A more pristine flux qubit with increased visibility and longer coherence times could display an even stronger violation of the macrorealist view. Our strict test of macrorealism provides evidence for a superposition of magnetic moments equivalent to several hundred thousand static electron spins pointing in opposite directions simultaneously. For further discussion on measures of macroscopicity, see Supplementary Note 3.

## Discussion
Note that the visibility of our measurement $V \approx 0.28$ is far below that required to find a violation of the LGI—this showcases the advantage of our scheme over standard tests of macrorealism. By eschewing LG's inequality, but upholding their methodology (as we have done here), improved bounds on macrorealist theories may be obtained more easily. This is in contrast with proposals that tend to increase the complexity both of the logical argument and of the experimental set-up; note that the requirements of a recent laboratory test of the LGI[22] extend to high visibility, partial-strength, non-demolition measurements of two-time correlators via entanglement with a coherent ancilla—requirements that are not necessary in our approach. Furthermore, our reasoning does not use any quantum mechanical assumptions, which, if relied on, can otherwise vitiate the refutation of macrorealism. With the experimental protocol duly simplified, the challenge now is to perform strict tests of macrorealism at much higher macroscopicities—a feat which should be possible as long as the classical disturbance of one's measurement is not too high.

## Methods
**Qubit design and fabrication.** Our three-junction flux qubit, fabricated using angled evaporation, is placed at the centre of a transmission line Josephson Bifurcation Amplifier (JBA) resonator where it is magnetically coupled (see Fig. 2). The Hamiltonian is

$$H = \frac{\epsilon}{2}(|\circlearrowright\rangle\langle\circlearrowright| - |\circlearrowleft\rangle\langle\circlearrowleft|) + \frac{\Delta}{2}(|\circlearrowright\rangle\langle\circlearrowleft| + |\circlearrowleft\rangle\langle\circlearrowright|) \qquad (8)$$

where $\Delta$ is the tunnelling energy and $\epsilon$ is the bias energy, and $|\circlearrowright\rangle$ and $|\circlearrowleft\rangle$ are states of definite supercurrent. The energy eigenstates are

$$|g\rangle = \sin x |\circlearrowright\rangle + \cos x |\circlearrowleft\rangle \qquad (9)$$

$$|e\rangle = \cos x |\circlearrowright\rangle - \sin x |\circlearrowleft\rangle \qquad (10)$$

where $2x = \arctan(\Delta/\epsilon)$. In our experiment, $\Delta/h = 2.75\,\mathrm{GHz}$ and $\epsilon/h \approx 2.90\,\mathrm{GHz}$ ($h$ is Planck's constant), hence approximately

$$|g\rangle = 0.37|\circlearrowleft\rangle + 0.93|\circlearrowright\rangle \qquad (11)$$

$$|e\rangle = 0.93|\circlearrowleft\rangle - 0.37|\circlearrowright\rangle. \qquad (12)$$

Our flux qubit has a resonance frequency of 4.0 GHz with a persistent current of $I_p = 170\,\mathrm{nA}$. It is chosen to have an average coherence time $T_2 \approx 10\,\mathrm{ns}$ (orders of magnitude below the best recorded time[4]) to show the advantages of our approach.

**Preparation, control and readout.** Initialization of our flux qubit (operating below 10 mK) is achieved by thermal relaxation. Microwave lines provide a mechanism for applying resonant qubit control pulses of fixed duration (2 ns), which rotate the qubit state by an angle proportional to the amplitude of the applied field. The second $S_2$ control pulse is applied 18 ns after the $S_1$ pulse. The effect of the fixed-duration pulses is controlled by modulating the power (see Fig. 3). To calibrate microwave power with the intended rotation angle $\theta$ we measured a $(2\theta, 0)$ sequence, and fitted a sinusoid to the data. The data agree qualitatively with a simple quantum mechanical model (see Supplementary Fig. 1).

Optionally, we apply a measurement pulse $O$ (at a frequency of 6.5 GHz with total length 12 ns with 4 ns rising time) to the JBA, turning it on at $t_2$ (between $S_1$ and $S_2$). This operation can be thought of as a measurement of the flux qubit, or (since we do not inspect the result) equivalently, as a completely dephasing operation $\mathcal{E}(\rho) = |\alpha|^2|g\rangle\langle g| + |\beta|^2|e\rangle\langle e|$. The qubit state is finally measured at $t_3$, again through coupling to the JBA in the standard manner[28–30].

**Statistics.** Variances in measured quantities were propagated according to elementary rules $\mathrm{Var}(d_\rho - d_g) = \mathrm{Var}(d_\rho) + \mathrm{Var}(d_g)$, giving $|d_\rho - d_g|/\sqrt{\mathrm{Var}(d_\rho - d_g)} \approx 84$. We conclude that our violation of macrorealism is of extremely high significance (See Fig. 2).

**Data availability.** All relevant data are available from the authors.

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

## Acknowledgements

We thank Erik Gauger for discussions. We acknowledge support from the MEXT Grant-in-Aid for Scientific Research on Innovative Areas 'Science of hybrid quantum systems', grant numbers 15H05870 & 15H05867.

## Author contributions

G. C. K. and M-C. Y. conceived and designed the experiment. G. C. K. and A. J. L. wrote the paper. K. K., H. T., H. Y. and S. S. performed the experiment. G. C. K., M-C. Y., Y. M. and W. J. M. modelled and refined the protocol. All authors contributed to refining the manuscript.

## Additional information

**Competing financial interests:** The authors declare no competing financial interests.

