## [Peer Review File · Nature Communications]

Reviewer #1 (Remarks to the Author):

The article contains two important results: a novel modification of the Leggett-Garg (LG) test, which replaces an inequality with an equality and thus significantly simplifies its experimental verification, and an experiment which confirmed the existence of coherent superposition of quite macroscopic currents in a superconducting flux qubit by over 77 standard deviations. The experiment also utilized the method proposed earlier by one of the authors to directly measure the effect of classical disturbances instead of making apriori assumptions.

The results provide a reliable, statistically robust and absolutely unequivocal proof that macroscopic superpositions (i.e., quantum mechanical description) survive up to this scale, which strongly supports the notion that a fundamental "Copenhagen" barrier between classical and quantum worlds does not exist, and that the solution to the measurement problem cannot be found on this way.

The paper is written clearly and logically and provides appropriate references to the earlier research. Its novelty and importance, especially in the context of modern push for quantum technologies and revitalization of the investigation of basics of quantum mechanics fully warrant its expedited publication.

I believe that the paper can be published in its current form, unless the authors decide to consider the following - admittedly nitpicking - remarks.

1) "quickly but adiabatically" (p.13, 2nd paragraph) - anybody involved in this particular research field will understand the meaning, but it may sound a bit strange to an average reader.

2) It would be useful to include in the supporting materials the definition of disconnectivity with some back-of-the-envelope formulas instead of a purely verbal discussion (p.7).

Reviewer #2 (Remarks to the Author):

The manuscript presents experimental test of macrorealism based on what the authors call ``streamlined Leggett-Garg approach' given by the Eq. 5 of the manuscript. In essence, this equation describes the basic feature of quantum mechanics that the measurements necessarily change the measured quantum state if done on a state with coherent superposition of components which correspond to different values of the measured observable. Results obtained with superconducting flux qubit confirm this feature of superposition states in contradiction to expectations from the macrorealism.

I see several problems with the manuscript which, for me, make it not suitable for publication in Nature Communications. The main problem is that the developments in experimental quantum information during the past 15 - 20 years have shown conclusively that, when it comes to the theory of qubits, no reasonable alternative exists to quantum mechanics. All of the experimental results published during this time, including those on superconducting qubits, were interpreted successfully in terms of quantum mechanics of open quantum systems. In this respect, the conclusion of an experimental work which simply refutes macrorealism in favor of quantum mechanics in a typical superconducting qubit cannot be viewed as something new and substantive. On the other hand, presented experiments themselves, by the present-day standards, are quite standard and simple, and do not constitute any progress over the previous results. While not very significant in comparison to the main problem, there is also a problem with presentation in the manuscript. The central element of the experiments done in this work is the measurement -induced perturbation O of the qubit state at time t_2 (Fig. 1). No information is given in the main text or supplementary information on what transformation of the qubit state this perturbation implements. Potential readers are referred to Ref. 27 for information on this. This makes it impossible to fully understand the details of the performed experiment from the manuscript.

Reviewer #3 (Remarks to the Author):

In this manuscript, the authors report on the results of an experiment on a superconducting flux qubit. While carefully characterizing the clumsiness of their measurements, they were able to demonstrate a violation of a non-disturbance condition, which follows from the assumptions made in a world view called macroscopic realism, thereby ruling out a classical explanation of their experiment.

Whether one can put into practice the thought experiment of Schrödinger's cat - preparing a quantum superposition of macroscopically distinct states - is one of the most exciting questions in quantum foundations. Experimental groups all around the world are trying to push the realm of quantum effects to increasingly macroscopic systems. In my opinion, the submitted manuscript constitutes a major step in this endeavour, not only because of the size of the achieved superposition states but mainly due to the methodological "cleanliness" of their approach. By performing control measurements, the authors were able to bound the disturbance effects of their measurements. Moreover, given the limited experimental visibility, they eschewed the Leggett-Garg inequalities and decided to use an alternative condition (called "no-signalling in time" or "quantum witness" in the literature) with a simplified measurement protocol. This clearly demonstrated the superiority of the latter condition, opening a route for future experiments with much higher macroscopicities.

The manuscript is written very well and very clearly. I recommend publishing this manuscript in Nature Communications with some minor revisions, taking into account the comments below:

1. In the abstract, the authors report a 77 standard deviation violation, while in the "Results" section they write about 84 standard deviations. This inconsistency must be explained or resolved. Moreover, the section "Statistics" should be expanded by at least a few more sentences. It is clear that the distance of the mean values of the distributions d_{ρ} and d_g is -0.063 , but it is unclear how the authors combine the two widths of these distributions such that the resulting number fits 77 (or 84?) times into the distance.

2. I believe Figure 2 can be improved. The sequence is g , e , ρ in panel (a) and in the caption, but it is different in panel (b). This is an unnecessary complication for the reader. I recommend to change the order in panel (b) and even include the labels " g ", " e ", and " ρ " into that panel. What is also confusing is that the figure and the caption state that three different $S1$ pulses were used for the three states, while referring to Figure 1. In Figure 1, however, $S1$ is only drawn for the ρ measurement, not the g or e measurement. I recommend to straighten this out and make everything more consistent.

3. In Figure 3, there seems to be a "mW" (Milliwatt) missing in the square bracket of the axis label, which is supposed to give the unit of the microwave power. While dBm is indeed defined with respect to a Milliwatt, it is itself dimensionless. Hence also $10^{(dBm/20)}$ is still dimensionless, and a power unit must be explicitly included.

4. This is a subtle point regarding the authors are saying in the abstract that their experiment rules out those theories which "would deny coherent superpositions of 170 nA currents over a 9 ns timescale". (A similar statement appears in the "Results" section.) I wonder what a Bohmian would say to this. Bohmian mechanics is certainly not ruled out by the experiment, as it makes the same predictions as standard quantum mechanics. Moreover, a Bohmian might very well say that - while the wavefunction indeed has superposition character - there are no superpositions of currents in this experiment. Every electron has a well-defined position and velocity at all times. Bohmian mechanics maintains "macrorealism per se" (MRPS) while violating "non-invasive measurability" (NIM), because a measurement changes the shape of the wavefunction and thus the subsequent evolution of the electrons. Experiments like the one presented can only put forward evidence against the conjunction of MRPS and NIM, not against MRPS alone.

5. Below Eq. (3) the authors write $\bar{3} = \bar{2} = \bar{1} = \bar{G}$ (with negation bars above the numbers) and say that the ensembles are "equivalent". I can agree with the wording but worry about the equation itself. The ensembles correspond, by definition, to the respective measurement arrangements, and they are clearly different. Maybe it is better to write $\langle \dots \rangle_{\bar{3}} = \langle \dots \rangle_{\bar{2}} = \langle \dots \rangle_{\bar{1}} = \langle \dots \rangle_{\bar{G}}$ (with negation bars above the numbers)?

6. Supplementary Information: Typos: "is, 2,5", "could then becomes", "optically".

I am confident that all the above points can be easily addressed by the authors.

End of report.

Reviewer #4 (Remarks to the Author):

In the manuscript entitled "A strict experimental test of macroscopic realism in a superconducting flux qubit" George C. Knee et al. explore the concept of macrorealism as originally formulated by Leggett and Garg in 1985. The authors begin deriving a "simplified" version of the Leggett-Garg inequalities, concluding the manuscript with an experimental test on a flux qubit.

In their pioneering work, Leggett and Garg derived a class of inequalities (the Leggett-Garg test) to test if quantum coherence in macroscopic objects can be realized in the laboratory. By assuming that:

1. A macroscopic system with at least two states is at all the times in one (and only one) of the available states (assumption of macrorealism);
2. The state of the system can be detected without disturbing it (assumption of non-invasive measurement),

Leggett and Garg derived a class of inequalities that any system (subject to 1 and 2) must obey. On the other side, if a series of measurements violate the Leggett-Garg test one or both of the hypothesis above must be abandoned. Leggett and Garg envisioned the method as a powerful way to investigate the existence of "macroscopic quantum coherence".

In their work, Knee and co-authors derive from the original Leggett-Garg inequalities a non-disturbance condition NDC (as named in the manuscript) used as witness for macroscopic realism. The advantages of NDC over the original Leggett-Garg (LG) inequalities are summarized as:

1. The first of three measurements required in LG can be omitted;
2. The second measurement can be a "blind measurement";
3. Only single-point expectation values are required instead of two-point correlation values.

As correctly reported by the authors (with appropriate references) similar witnesses have been derived by others. For such a reason I suggest/recommend the authors to describe in more details how the proposed witness differs from others. Moreover, some explanation is required on the non-invasive nature of the measurement. Is this required, as it is in the original Leggett-Garg test, or not?

EXPERIMENTAL PART

I found the experimental part not clear and very concise in the exposition of the work. I would like to see more data supporting the experimental evidence, a description of the calibration procedures, as well experimental details (chip layout, connectivity, setup, etc.).

Other major considerations:

1. Figure 3 has very little explanation. The whole manipulation sequence is composed by two pulses (generating S_1 and S_2) with a readout pulse in between. Plots in the figure are drawn versus microwave power. If the two pulses have the same duration (2ns), how are they calibrated to provide $(2\theta, \theta)$ in the final figure? How is the qubit phase evolution between the two pulses considered? What is the reason for the drift observed in the oscillations (clearly visible in the bottom plot)? How is the drift considered and handled in the elaboration of the data?

2. Figure 2 is not very informative. Is there a reason the author measured the witness for only one state of the qubit (excluding the calibrations in the "controlling experiments")? I think that a plot of the witness versus the Rabi rotation angle in the whole Bloch sphere (0-2 π) would be more informative than the presented single point.

3. The operation O (observation) at t_2 is defined as a "blind measurement". According to reference [26] this is a measurement operation where the outcome is discarded. Knee and co-authors use a "measurement pulse" simulating the effect of the measurement. What are the requirements for this operation (non-invasive, weak, QND,...) and how is experimentally proved that the requirements are satisfied in the final sequence? This type of experimental investigation is not reported in the manuscript.

As a final remark, although I found the paper focused on a clear goal in an interesting and active topic, I do not feel like it represents a large/substantial advance over previous works either from a theoretical or experimental point of view (as an example, a more stringent violation of the macrorealism witnesses in superconducting qubit has been reported by J.P. Groen et al., "Partial-measurement backaction and non classical weak values in a superconducting circuit", Phys. Rev. Lett. 111, 090506 (2013)).

Point-by-Point reply to the Reviewers NCOMMS-16-04814-T

Reviewer	Reply / Changes
Reviewer #1: The paper is written clearly and logically and provides appropriate references to the earlier research. Its novelty and importance, especially in the context of modern push for quantum technologies and revitalization of the investigation of basics of quantum mechanics fully warrant its expedited publication. I believe that the paper can be published in its current form, unless the authors decide to consider the following - admittedly nitpicking - remarks. 1) "quickly but adiabatically" (p.13, 2nd paragraph) - anybody involved in this particular research field will understand the meaning, but it may sound a bit strange to an average reader. 2) It would be useful to include in the supporting materials the definition of disconnectivity with some back-of-the-envelope formulas instead of a purely verbal discussion (p.7).	We thank the referee for their encouraging remarks. We would like to respond to the points that they raise: 1.) We have removed this sentence.2.) Unfortunately, it is difficult to be more quantitative here – although equation (12) is indeed an explicit calculation performed by Korsbakken et al. We now include the following sentence in order to be more informative: The disconnectivity should measure the number of particles that behave differently in the two branches of the superposition. As with entanglement measures, with which the disconnectivity shares an affinity, there are a variety of ways of doing so – an example is shown below.

Reviewer #2:

The main problem is that the developments in experimental quantum information during the past 15 - 20 years have shown conclusively that, when it comes to the theory of qubits, no reasonable alternative exists to quantum mechanics. All of the experimental results published during this time, including those on superconducting qubits, were interpreted successfully in terms of quantum mechanics of open quantum systems.

On the other hand, presented experiments themselves, by the present-day standards, are quite standard and simple, and do not constitute any progress over the previous results.

The central element of the experiments done in this work is the measurement - induced perturbation O of the qubit state at time t_2 (Fig. 1). No information is given in the main text or supplementary information on what transformation of the qubit state this perturbation implements. Potential readers are referred to Ref. 27 for information on this. This makes it impossible to fully understand the details of the performed experiment from the manuscript.

*No amount of experimental evidence can rule a theory **in**. It is therefore not our aim to show that quantum mechanics is correct, or even to show that our experiment is consistent with quantum mechanics.*

Rather, it is the idea that quantum mechanics could break down at some critical level of macroscopicity that we wish to investigate. Such a view is gaining increased popularity, particularly with those in the quantum gravity community. Our results show that if there is such a breakdown of quantum mechanics, it must occur at a macroscopicity somewhat higher than that exhibited by our flux qubit. We would admit that this is hardly a surprise – but emphasize that it is the streamlined protocol and methodology that really constitutes the step forward in this field (as the other reviewers recognize).

Referee 3 calls our improved approach to falsifying macrorealism a major step in one of the most exciting questions in quantum foundations, mainly due to the methodological cleanliness of our approach. It is this that should spur on future tests at much higher macroscopicities. Those tests have the potential to decide on the future of physics.

We added more detail to the description of O , including a full description of the transformation it implements:

This operation can be thought of as a measurement of the flux qubit, or (since we do not inspect the result) equivalently, as a completely dephasing operation $E(\rho) = |\alpha|^2|g\rangle\langle g| + |\beta|^2|e\rangle\langle e|$.

Reviewer #3:

Whether one can put into practice the thought experiment of Schrödinger's cat - preparing a quantum superposition of macroscopically distinct states - is one of the most exciting questions in quantum foundations. Experimental groups all around the world are trying to push the realm of quantum effects to increasingly macroscopic systems. In my opinion, the submitted manuscript constitutes a major step in this endeavour, not only because of the size of the achieved superposition states but mainly due to the methodological "cleanliness" of their approach.

The manuscript is written very well and very clearly. I recommend publishing this manuscript in Nature Communications with some minor revisions, taking into account the comments below:

1. In the abstract, the authors report a 77 standard deviation violation, while in the "Results" section they write about 84 standard deviations. This inconsistency must be explained or resolved. Moreover, the section "Statistics" should be expanded by at least a few more sentences. It is clear that the distance of the mean values of the distributions d_{ρ} and d_g is - 0.063, but it is unclear how the authors combine the two widths of these distributions such that the resulting number fits 77 (or 84?) times into the distance.

2. I believe Figure 2 can be improved. The sequence is g, e, rho in panel (a) and in the caption, but it is different in panel (b). This is an unnecessary complication for the reader. I

We thank the reviewer for their encouraging comments.

We respond to the comments below:

- 1.) *We apologize for this slip. In fact we were able to gather more and more data to get to 84 sd, but made a typographical error in updating our manuscript. This is now fixed. We have also explained how the uncertainty of d_{ρ} and d_g is propagated to the uncertainty in the difference: Variances in measured quantities were propagated according to elementary rules $\text{Var}(d_{\rho} - d_g) = \text{Var}(d_{\rho}) + \text{Var}(d_g)$, giving $|d_{\rho} - d_g| / \text{Var}(d_{\rho} - d_g) \approx 84$.*
- 2.) *Thank you for this suggestion. We have made the changes to Figure 2 as per the suggestions. We also changed Figure 1, it is now consistent.*

recommend to change the order in panel (b) and even include the labels "g", "e", and "rho" into that panel. What is also confusing is that the figure and the caption state that three different S1 pulses were used for the three states, while referring to Figure 1. In Figure 1, however, S1 is only drawn for the rho measurement, not the g or e measurement. I recommend to straighten this out and make everything more consistent.

3. In Figure 3, there seems to be a "mW" (Milliwatt) missing in the square bracket of the axis label, which is supposed to give the unit of the microwave power. While dBm is indeed defined with respect to a Milliwatt, it is itself dimensionless. Hence also $10^{(dBm/20)}$ is still dimensionless, and a power unit must be explicitly included.

4. This is a subtle point regarding the authors are saying in the abstract that their experiment rules out those theories which "would deny coherent superpositions of 170 nA currents over a 9 ns timescale". (A similar statement appears in the "Results" section.) I wonder what a Bohmian would say to this. Bohmian mechanics is certainly not ruled out by the experiment, as it makes the same predictions as standard quantum mechanics. Moreover, a Bohmian might very well say that - while the wavefunction indeed has superposition character - there are no superpositions of currents in this experiment. Every electron has a well-defined position and velocity at all times. Bohmian mechanics maintains "macrorealism per se" (MRPS) while violating "non-invasive measurability" (NIM), because a measurement changes the shape of the

3.) *Thank you for spotting this. We corrected the units.*

4.) *The reviewer is quite right to raise Bohmian mechanics as a hidden variable interpretation. As he/she says, this interpretation can reproduce all the predictions of quantum theory yet seems to subscribe to MRPS. It is a subtle point that is difficult to full justice to in a short abstract – however, we feel that a sufficiently generous understanding of ‘coherent superposition’ would put all readers (even the Bohmians at ease). By coherent superposition, we mean a ‘state’ (taken in the most general sense) that violates our NDC. That might be thought of as an actual superposition of currents, or even a well-defined current along with the requisite guidance wave or quantum potential that is necessary to exhibit interference. In short, one can frame decoherence (be it environmental or of the GRWP type) within the theory of Bohmian mechanics – and then*

wavefunction and thus the subsequent evolution of the electrons.

Experiments like the one presented can only put forward evidence against the conjunction of MRPS and NIM, not against MRPS alone.

5. Below Eq. (3) the authors write $\bar{3} = \bar{2} = \bar{1} = \bar{G}$ (with negation bars above the numbers) and say that the ensembles are "equivalent". I can agree with the wording but worry about the equation itself. The ensembles correspond, by definition, to the respective measurement arrangements, and they are clearly different. Maybe it is better to write $\langle \dots \rangle_{\bar{3}} = \langle \dots \rangle_{\bar{2}} = \langle \dots \rangle_{\bar{1}} = \langle \dots \rangle_{\bar{G}}$ (with negation bars above the numbers)?

6. Supplementary Information: Typos: "is, 2,5", "could then becomes", "optically".

I am confident that all the above points can be easily addressed by the authors.

there is still a useful distinction between coherent and noncoherent states (and of course there is arguably a very different metaphysical or ontological distinction compared with more orthodox interpretations of quantum theory). Under that framework, we believe our sentence to make sense.

5.) *We used the referee's suggestion.*

6.) *Thank you, we fixed these.*

Reviewer #4

As correctly reported by the authors (with appropriate references) similar witnesses have been derived by others. For such a reason I suggest/recommend the authors to describe in more details how the proposed witness differs from others. Moreover, some explanation is required on the non-invasive nature of the measurement. Is this required, as it is in the original Leggett-Garg test, or not?

We thank the reviewer for their insightful comments. We would like to reply to each point raised:

Firstly, the NDC does not differ in essence from the other witnesses given in the reference, (aside sometimes, from a factor of 2). A new feature that we introduce in our work is to relax the need for the operation at t_2 to be interpreted as a measurement. This is summarised in the new sentence:

Here we suggest, however, that the requirements for the measurement at t_2 are very minimal – it will be clear shortly that it need not even be a measurement at all, but some generalised operation O about which we need not assume anything. All pertinent properties of O are to be obtained through experiment.

*As we state, the non-invasive nature of the measurement is indeed important and has been motivated in a number of ways in recent years. It is required here, too. Our approach here is to **measure** the invasiveness of the measurement. To reflect this, we have added the sentences:*

The ever-present possibility of a clumsy measurement procedure at t_2 giving rise to a violation, rather than any inherent quantum effect, is as important here as ever. The issue has so far only been addressed with *a priori* arguments – those from the use of null result measurements, weak measurements or the use of an additional ‘stationarity’ assumption. By contrast, Leggett and later Wilde and Mizel have proposed that the problem can be attacked experimentally. This is precisely the

EXPERIMENTAL PART

I found the experimental part not clear and very concise in the exposition of the work. I would like to see more data supporting the experimental evidence, a description of the calibration procedures, as well experimental details (chip layout, connectivity, setup, etc.).

Other major considerations:

1. Figure 3 has very little explanation. The whole manipulation sequence is composed by two pulses (generating S1 and S2) with a readout pulse in between. Plots in the figure are drawn versus microwave power. If the two pulses have the same duration (2ns), how are they calibrated to provide $(2\theta, \theta)$ in the final figure? How is the qubit phase evolution between the two pulses considered? What is the reason for the drift observed in the oscillations (clearly visible in the bottom plot)? How is the drift considered and handled in the elaboration of the data?

2. Figure 2 is not very informative. Is there a reason the author measured the witness for only one state of the qubit (excluding the calibrations in the "controlling experiments")? I think that a plot of the witness versus the Rabi rotation angle in the whole Bloch sphere ($0-2\pi$) would be more informative than the presented single point.

approach we adopt here: The classical disturbance of a measurement (which we define shortly) may be determined in a control experiment, rather than assumed zero.

Experimental part: We have included more detail on the experimental procedures, in particular the addition of an electron micrograph and schematic of our device to Figure 2.

1. We added the following sentences:

In order to calibrate microwave power with the intended rotation angle θ , we measured a $(2\theta, 0)$ sequence, and fitted a sinusoid to the data. The slight drift is due to slowly varying fluctuations in an externally applied magnetic field. This is measured and corrected for at the start of each measurement sequence. The time evolution of the qubit between the control pulses is on such a short timescale that it may safely be ignored.

2. We only need one point to find a violation of macrorealism, so finding more points is superfluous to our requirements. Further, due to the design of our qubit and associated circuitry, the data was gathered in experiments where the pulse angles were $0, \theta$; θ, θ ; and $2\theta, \theta$. It is therefore not possible to extract π, θ and $\pi/2, \theta$ without a significant number of new experiments. As stated above, this is actually more than is required.

3. The operation O (observation) at t_2 is defined as a "blind measurement". According to reference [26] this is a measurement operation where the outcome is discarded. Knee and co-authors use a "measurement pulse" simulating the effect of the measurement. What are the requirements for this operation (non-invasive, weak, QND,...) and how is it experimentally proved that the requirements are satisfied in the final sequence? This type of experimental investigation is not reported in the manuscript.

As a final remark, although I found the paper focused on a clear goal in an interesting and active topic, I do not feel like it represents a large/substantial advance over previous works either from a theoretical or experimental point of view (as an example, a more stringent violation of the macrorealism witnesses in superconducting qubit has been reported by J.P. Groen et al., "Partial-measurement backaction and non-classical weak values in a superconducting circuit", Phys. Rev. Lett. 111, 090506 (2013)).

3. The requirements of the operation O are very weak indeed. The invasiveness of the operation is explicitly measured (this is d_g and d_e). The invasiveness must not be too large, otherwise the eq 7 cannot be violated. The operation need not satisfy any other requirements. See new sentence (above).

We would question whether the PRL referenced by the referee is a more stringent violation of macrorealism. That paper contains a more complicated test that relies on the introduction of new assumptions, particularly concerning the effect of weak measurements. Moreover, it is not particularly novel – in fact it is really performing the experiment of Goggin et al (ref 21), albeit in a superconducting system. By contrast, our paper breaks genuinely new ground by refining and improving the protocol before employing it. Crucially, we have been the first to implement an ancillary test of the invasiveness of the measurements – a step not taken in any previous work, including the PRL in question. The control experiments are important in addressing loopholes in the argument. We include a reference to this PRL so the readers can see our advance for themselves.

Our paper is undoubtedly different to all previous experimental tests of macrorealism, and we would argue far 'cleaner' (as referee 1 puts it). The theoretical advances, which are supported by an experimental test, are plain to see – rather than making tests of macrorealism more and more complicated and difficult, one can employ our very simple test and

learn at least the same amount about the validity of macrorealism. We added the sentence:

This is contrast with proposals that tend to *increase* the complexity both of the logical argument and of the experimental setup – note that the experimental requirements of a recent test of the LGI²² extend to high visibility, partial-strength, non-demolition measurements of two-time correlators via entanglement with a coherent ancilla; requirements that are not necessary in our approach. Furthermore, our reasoning does not rely on any quantum mechanical assumptions, which, if relied upon, can otherwise vitiate the refutation of macrorealism.

Reviewer #3 (Remarks to the Author):

This is my second report for the manuscript "A strict experimental test of macroscopic realism in a superconducting flux qubit".

The authors have sufficiently addressed the points I had raised in my first report. I therefore recommend publication.

Remarks:

Newly introduced typos: caption 1: "only the a maximally"; caption 2: "bound with at high degree".

I have been asked by the editor about my opinion on the extent to which I feel the authors have satisfactorily replied to the comments of Reviewer #2. The criticism of reviewer #2 had three main points: First, he/she criticised that all existing experiments can be interpreted in terms of quantum mechanics. Here, I share the authors' opinion that the merit of these experiments is to prove that a potential breakdown of quantum mechanics must be beyond their macroscopicity. Experiments pushing quantum effects to more and more macroscopic systems are thus a very worthwhile enterprise. Second, he/she criticised that these experiments "do not constitute any progress over the previous results". Here, I maintain my view that the present manuscript does indeed constitute a significant progress compared to earlier experiments, mainly due its clean methodology. Third, he/she criticised that that operation O is not explained well enough. Here, I believe the authors have sufficiently improved the description of this operation in the manuscript. In total, I feel that the criticism of reviewer #2 has been addressed successfully.

End of report.

Reviewer #4 (Remarks to the Author):

I have appreciated the work done by the authors to clarify my points. I found the manuscript improved from the previous version, but I still feel like the paper is incomplete from an experimental point of view.

The authors claim that the "slight" drift in the Rabi oscillations presented in figure 3 is due to slowly varying fluctuations in an externally applied magnetic field. This raises two points. First, the drift is not small considering that in figure 3c is clearly visible a drift of approximately the 50% of the oscillation visibility in only one period. Second, holding the hypothesis of varying fluctuation of the external magnetic field I expect a drift in the oscillation that is "time-dependent", leading to unpredictable starting condition and drift for the Rabi oscillation. If this is the case, the oscillations presented in figure 3 are specific of a defined measurement in time, and cannot be reproduced.

This opens the question of "verification" of the presented results. The main claim of violation of the inequality is based on a single measurement (excluding the control experiments). The disturbance parameter of a particular state ρ , calculated as difference of the two curves presented in figure 3b in correspondence of the vertical dotted line, is compared with the disturbance calculated in figure 3a and 3c (results presented in figure 2). In the presence of a fluctuating external magnetic field, the oscillations in Figure 3 are subject to drifting. How can the author prove that the disturbance measured is the pure effect of macrorealism and not a simple fluctuation?

My suggestion to plot the witness versus the Rabi rotation angle (instead of a single point) is based on the fact that the reader can "validate" the main claim of violation (I would not consider it "superfluous to the experiment"). In fact for the $|0\rangle$ and $|1\rangle$ states one expect to measure the same disturbance of the control experiment, with a violation of the inequality that increases with the Rabi rotation angle up to a maximum, to decrease back toward the disturbance of the control experiment.

Point-by-Point reply to the Reviewers NCOMMS-16-04814A

Reviewer	Reply / Changes
Reviewer #3: The authors have sufficiently addressed the points I had raised in my first report. I therefore recommend publication. Remarks: Newly introduced typos: caption 1: "only the a maximally"; caption 2: "bound with at high degree". I feel that the criticism of reviewer #2 has been addressed successfully.	We thank the referee for their support and for pointing out the typos. These have now been fixed.

Reviewer #4:

I have appreciated the work done by the authors to clarify my points. I found the manuscript improved from the previous version, but I still feel like the paper is incomplete from an experimental point of view.

The authors claim that the "slight" drift in the Rabi oscillations presented in figure 3 is due to slowly varying fluctuations in an externally applied magnetic field. This raises two points. First, the drift is not small considering that in figure 3c is clearly visible a drift of approximately the 50% of the oscillation visibility in only one period. Second, holding the hypothesis of varying fluctuation of the external magnetic field I expect a drift in the oscillation that is "time-dependent", leading to unpredictable starting condition and drift for the Rabi oscillation. If this is the case, the oscillations presented in figure 3 are specific of a defined measurement in time, and cannot be reproduced.

This opens the question of "verification" of the presented results. The main claim of violation of the inequality is based on a single measurement (excluding the control experiments). The disturbance parameter of a particular state ρ , calculated as difference of the two curves presented in figure 3b in correspondence of the vertical dotted line, is compared with the disturbance calculated in figure 3a and 3c (results presented in figure 2). In the presence of a fluctuating external magnetic field, the oscillations in Figure 3 are subject to drifting. How can the author prove that the disturbance measured is the pure effect of macrorealism and not a simple fluctuation?

*We apologise for not being clearer here: Figure 3c is not a standard Rabi oscillation, since there are *two* pseudo-spin rotations (interrupted by dephasing). As we now show in the supplementary material, the ideal quantum mechanical predictions show qualitatively very similar curves. The shape of the curves is not characteristic of substantial drift, but merely of the beating of the two rotation frequencies. Although not obvious from the Figure, the evolution remains periodic (as can be seen by the new analytical formula $\langle Q3 \rangle = -\cos(2x)\cos(x)$ in the supplementary material). There is a highly predictable initial condition: The reproducibility of our experiment is extremely good. We performed an enormous amount of experiments and extracted a violation of over 80 standard deviations. So the likelihood that our data are simply the result of a fluctuation is astronomically small.*

We appreciate that plotting the witness against the Rabi rotation angle would allow the reader to verify the QM predictions at a glance, there are two reasons we chose not to do this. The first is that, as we have explained in detail, our goal is not to show QM is correct but that MR is incorrect. Such is the essence of the scientific method. This theory-neutral approach to experiments has a long history in the Bell inequality experiments, where they choose the optimal measurement settings and use all resources to increase the statistical confidence. Our second reason is related: not only would scanning the Rabi angle consume many more experimental resources (to the detriment of statistical confidence), as we mentioned previously it so happens

My suggestion to plot the witness versus the Rabi rotation angle (instead of a single point) is based on the fact that the reader can "validate" the main claim of violation (I would not consider it "superfluous to the experiment"). In fact for the $|0\rangle$ and $|1\rangle$ states one expects to measure the same disturbance of the control experiment, with a violation of the inequality that increases with the Rabi rotation angle up to a maximum, to decrease back toward the disturbance of the control experiment.

to be quite inconvenient from an experimental point of view. To do so would require essentially a brand new experiment and many months of work. We do not believe that such a step would improve our paper enough to warrant it. Furthermore, the qualitative agreement we now show between theory (supplementary figure 1) and experiment (Figure 3) will indeed allow the reader to "validate" that our qubit is well described by quantum mechanics.

Please see new section in the supplementary material.

Reviewer #4 (Remarks to the Author):

I appreciate the new section "Ideal quantum mechanical predictions in Figure 3 of the main text" in the supplementary material. This new part addresses and clarify my observation on the anomalous Rabi oscillation presented in Figure 3 of the main text.

I therefore recommend publication in Nature Communications.